# Integrated Transcriptome and Metabolome Analysis Revealed the Causal Agent of Primary Bud Necrosis in ‘Summer Black’ Grape

**DOI:** 10.3390/ijms241210410

**Published:** 2023-06-20

**Authors:** Shaogang Fan, Yanshuai Xu, Miao Bai, Feixiong Luo, Jun Yu, Guoshun Yang

**Affiliations:** College of Horticulture, Hunan Agricultural University, Changsha 410128, China

**Keywords:** grapevine, primary bud necrosis, protein misfolding, reactive oxygen species, transcriptome, tissue browning, *VvP231*

## Abstract

Primary bud necrosis of grape buds is a physiological disorder that leads to decreased berry yield and has a catastrophic impact on the double cropping system in sub-tropical areas. The pathogenic mechanisms and potential solutions remain unknown. In this study, the progression and irreversibility patterns of primary bud necrosis in ‘Summer Black’ were examined via staining and transmission electron microscopy observation. Primary bud necrosis was initiated at 60 days after bud break and was characterized by plasmolysis, mitochondrial swelling, and severe damage to other organelles. To reveal the underlying regulatory networks, winter buds were collected during primary bud necrosis progression for integrated transcriptome and metabolome analysis. The accumulation of reactive oxygen species and subsequent signaling cascades disrupted the regulation systems for cellular protein quality. ROS cascade reactions were related to mitochondrial stress that can lead to mitochondrial dysfunction, lipid peroxidation causing damage to membrane structure, and endoplasmic reticulum stress leading to misfolded protein aggregates. All these factors ultimately resulted in primary bud necrosis. Visible tissue browning was associated with the oxidation and decreased levels of flavonoids during primary bud necrosis, while the products of polyunsaturated fatty acids and stilbenes exhibited an increasing trend, leading to a shift in carbon flow from flavonoids to stilbene. Increased ethylene may be closely related to primary bud necrosis, while auxin accelerated cell growth and alleviated necrosis by co-chaperone VvP23-regulated redistribution of auxin in meristem cells. Altogether, this study provides important clues for further study on primary bud necrosis.

## 1. Introduction

Two types of grapevine buds are formed in the leaf axils, the winter bud and prompt bud. The prompt bud is a genuine axillary bud that arises from the axil of a foliage leaf on a growing shoot and develops into the lateral shoot while the main shoot continues to grow. The winter bud is a compound bud that is composed of primary and secondary buds, and these buds remain dormant until the next growing season. The primary bud originates from the prophyll of lateral shoots and is typically located at the center, and it is the largest bud in the compound bud. The primary bud later develops into the fruiting vine next season. The secondary latent buds arise in the prophylls of the primary bud and are located at the axils of the primary bud, which are usually not necessary for fruiting unless the primary bud dies [1,2].

Primary bud necrosis (PBN) on grape buds is a physiological disorder that causes the primary bud to die while the secondary buds remain healthy [3,4,5]. Although the secondary buds can produce fruitful vines, it is widely recognized that PBN leads to decreased berry yield since secondary shoots are typically less fruitful than the primary shoots [6]. Moreover, PBN exerts a catastrophic impact on the double cropping system and hinders the feasibility of the two-crop-a-year cultivation technique in South China, which is an important table grape production region. PBN makes the double cropping system more difficult and infeasible to achieve additional economic benefits from winter cropping harvests [7]. 

Histological analysis has shown that PBN is typically associated with bud development, with necrotic cells observed around 60 days after bud break (DABB) until the onset of bud dormancy, although the fundamental causes are not yet clear [4,5,8]. Studies have suggested that susceptible cultivars and viticultural stresses affecting bud growth may be responsible for the PBN [5]. Susceptible cultivars include Kyoho [9], Queen of Vineyard [10], Riesling [3], Flame Seedless, Thompson Seedless [4], and Shiraz [5]. Stresses, such as shoot vigor [11,12,13], canopy shading [8,14,15], and gibberellin (GA) level, positively regulate the incidence of PBN [16,17]. PBN is more severe in subtropical viticulture regions located in mid-to-low-latitude areas or warm winter areas, such as Adelaide in Australia (34° S) [11], Três Corações in Brazil (21° S) [18], San Bernardo in Chile [14], Israel [10,16] (30° N), and U.S. states, such as California and Virginia [3,4]. Our observation shows that the incidence of PBN is more severe in South China than in North China, including Hunan, Sichuan, Guangxi, and Yunnan provinces. However, there is limited research on PBN in traditional temperate cultivation areas. Grapevine PBN like bud necrosis observed in pear trees is described as flower bud necrosis (FBN) [19]. For example, pears grown in low-altitude areas with high average annual temperatures, such as pear cultivar Shinko, have a fast flower bud differentiation and a high rate of flower bud death, while in high-altitude areas, the ambient conditions are different, with slower flower bud differentiation and a lower rate of flower bud death [20]. However, researchers detected no clear association between climatic conditions and pear FBN in Uruguay; in contrast, they found FBN is caused by *Pseudomonas* bacteria [21]. Based on morphological and anatomical observation and previous pathology experiments, we tend to believe that grapevine bud necrosis is more like non-infectious bud failure (NBF) in almonds, which is associated with DNA methylation [22,23], rather than diseases caused by the pathogenic bacteria, such as *Alternaria alternata* [24]. 

In conclusion, although bud necrosis is a widespread issue in grapevine cultivation, the causative factors, underlying mechanisms, and potential solutions are yet to be fully understood. Reactive oxygen species (ROS) are considered to be early signals of programmed cell death and cell necrosis, causing damage to cellular components [25,26,27]. However, the role of ROS in the mechanisms of PBN has not been reported yet. Integrated transcriptome and metabolome analysis offers a comprehensive description of cellular metabolic phenotypes, while differential metabolite information accurately reflects the true biological phenotype by filtering out non-functional gene and protein expression changes. This integrated approach enables the exploration of intricate biological processes in fruit trees, such as PBN. In this study, we investigated the development and severity of PBN in grapevine cv. Summer Black. We aimed to explore the regulatory networks involved in the development of PBN and reveal the underlying mechanism through employing transcriptomics and metabolomics technologies.

## 2. Results

### 2.1. Primary Bud Necrosis Progression in ‘Summer Black’

Phenological observations of ‘Summer Black’ grape found that the bud break occurred around 20 March every year, and PBN began on 22 May, which was 63 DABB, as observed through the transverse sections of buds in the field. PBN was concentrated and regularly occurred in the third to sixth nodes, lasting more than 7 days, with complete death occurring at 70 DABB. The PBN process was divided into three stages: before PBN (BN), during PBN (PN), and absolute PBN (AN). Healthy primary buds appeared green with a high water content in BN, while the tissue encountered browning and easily crumbled during dissection in PN. In AN, the tissue rotted into black, muddy, or black-brown fragmented tissue (Figure 1a). Necrosis at the base of the first and second nodes was relatively delayed by more than one week.

Necrosis in the form of browning occurred in the prophylls, bract, inflorescence primordia, and shoot primordia during PN, but it is difficult to detect at early stages. Cell death in the bud tissue occurred before it could be visually observed. Staining appeared as dark blue spots in PN and AN (Figure 1b). A clear boundary appeared between the bud cushion and the bud base, where cells died in scattered form and appeared as deep-blue spots (Figure 1c). Necrosis stopped at the bud base and, ordinarily, did not spread to the lateral buds. A hollow cavity formed at the site of the bud cushion, where cells break down, the bud cushion and bud base separate, with a clear boundary between them.

**Figure 1 ijms-24-10410-f001:**
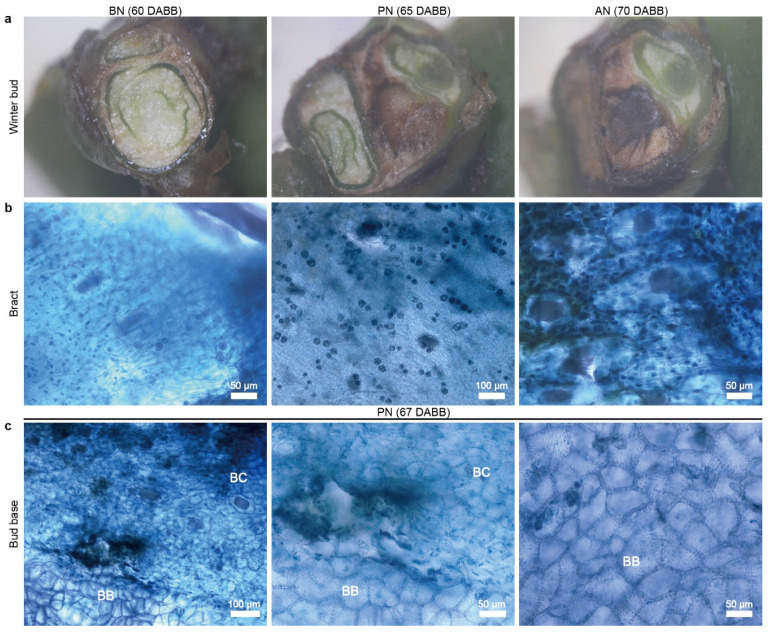
The primary bud necrosis progression in grapevine ‘Summer Black’ at sixth node. (**a**) Cross-section of grape winter bud. (**b**) Bract staining using trypan blue reveals the process of primary bud necrosis from 60 to 70 DABB. (**c**) Primary bud necrosis occurs in bud base tissues at 67 DABB. DABB, days after bud break; PBN, primary bud necrosis; BN, before PBN; PN, during PBN; AN, absolute PBN; BC, bud cushion; BB, base of compound bud.

The primary bud at the sixth node had a fully developed inflorescence primordia at 50 DABB (Figure 2a). The peripheral part of the bud tissue was stained blue with 4,6-diamidino-2-phenylindole (DAPI) for the nuclei of living cells, while the bud axis beneath the inflorescence primordia was stained red with tetramethyl-rhodamine-5-dUTP (TMR) for degraded DNA within the nuclei, indicating that genome breakage occurred. The degree of necrosis increased along the bud axis upward after 60 to 70 DABB (Figure 2b,c). Only the cells at the top of the inflorescence primordia retained genome integrity, and there was less DAPI staining observed at the bud axis, indicating a lower number of viable cells (Figure 2d). Dead cells were easily detached during the dewaxing process due to cell necrosis, leaving only live cells in the tissue at the top of the inflorescence. The PBN process was found to develop from the bottom of the bud axis to the top of the inflorescence primordia or shoot primordia. 

The bud tissues were dissected into bracts, inflorescence primordia or shoot primordia, and the bud cushions and then were examined using transmission electron microscopy (TEM). Subcellular observations revealed that samples had a comparatively intact structure in BN, with clear intercellular connections, abundant cytoplasm and organelles, and clear primary wall and secondary wall structures. However, a small proportion of cells underwent plasmolysis, the mitochondria showed mild swelling, and the chloroplasts degraded, leading to disturbance in the cytoplasm (Figure 3). The healthy bract cells exhibited fewer organelles, and protein aggregation was observed in structurally intact cells of the bracts. 

In PN, the cell wall structure of all tested tissues was abnormal, with destruction of fiber structures and deeper electron density of the cell wall, resulting in an increased cell wall thickness and decreased permeability. Mitochondria were severely swollen, and chloroplast and the Golgi apparatus structure were more severely damaged with prominent separation of the cell walls. Damaged proteins tend to aggregate to form protein aggregates, and protein condensation was observed in the apical primordia and bract tissue cells. Additionally, white starch grains and lipid droplet structures were visible (Figure 3a). The cells closely adjacent to the necrotic cells showed significant membrane separation and expansion of the endoplasmic reticulum (ER) into vesicular structures (Figure 3b). Cells of the bract became severely deformed with disorganized cellulose, holes in the cell walls, and disordered cytoplasm flowing out of the cells, leading to the complete destruction of the cell wall structure (Figure 3c).

**Figure 3 ijms-24-10410-f003:**
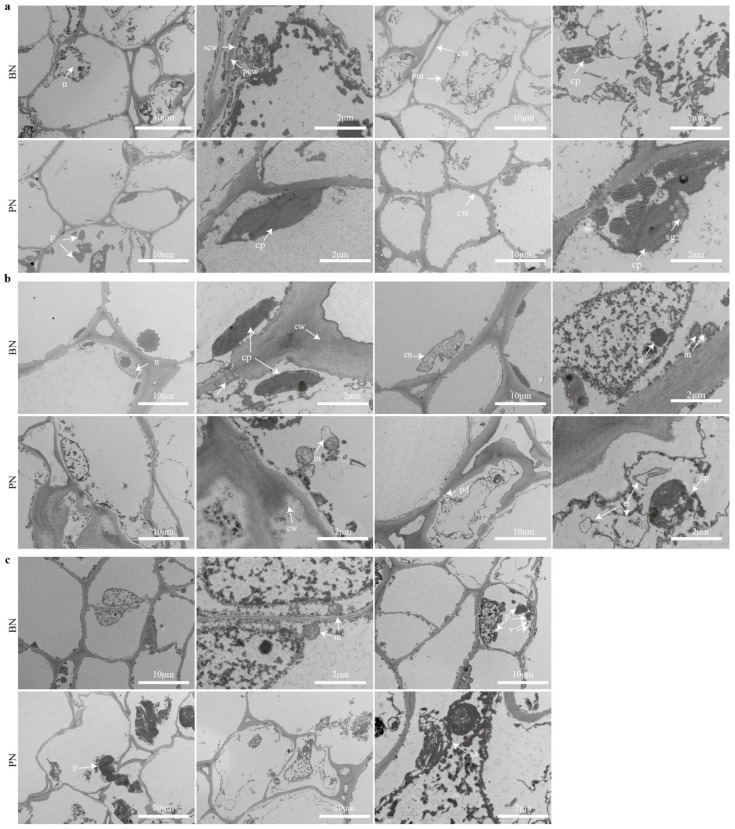
Ultrastructural changes in primary cells during primary bud necrosis. Buds sampled from BN and PN were dissected into bracts, apical meristem, and bud cushions. (**a**) Apical meristems of primary bud. (**b**) Bud cushion (**c**) Bract. cn, cell nucleus; cp, chloroplast; er, endoplasmic reticulum; ga, Golgi apparatus; m, mitochondrion; n, nucleolus; p, protein aggregate; pcw, primary cell wall; pm, plasma membrane; scw, secondary cell wall; sg, starch granule; v, vacuole.

### 2.2. The Changes in Hormone and H_2_O_2_ Levels during PBN Progression in ‘Summer Black’

To investigate the role of ROS in bud necrosis, the apical meristem and bract of primary buds in PN were stained with fluorescent H_2_O_2_ probe hydroxyphenyl fluorescein (HPF), and conventional 3,3′-diaminobenzidine (DAB) staining was used as supporting evidence. In PN, a significant accumulation of ROS was observed in the bract and apical meristem, particularly in the primary bud axis and bud cushion, as indicated by the intense signals of H_2_O_2_ (Figure 4a). This suggests that a large amount of ROS accumulates during the process of bud necrosis, and ROS is closely associated with PBN. The H_2_O_2_ content was 7.55 μmol·g^−1^ fresh weight (FW) during BN, with the lowest content being 6.81 μmol·g^−1^ FW during PN and the highest content during AN (Figure 4b). These results suggest that ROS may be the inducer of cell death.

According to the results of phytohormone content detection, almost all hormone levels were significantly increased simultaneously during PBN, indicating a vigorous growth activity of the bud. It was observed that the content of auxin indole-3-acetic acid (IAA) increased three-fold. Additionally, the levels of indolic derivatives, such as indole-3-lactic acid (ILA) and indole-3-carboxaldehyde (ICAld), doubled, while indole-3-carboxylic acid (ICA) showed a 15-fold increase after the onset of bud necrosis (Figure 4c). Furthermore, the products of IAA oxidative metabolism, including 2-oxindole-3-acetic acid (oxIAA), indole-3-acetyl glutamic acid (IAA-Glu), and indole-3-acetyl-L-aspartic acid (IAA-Asp), exhibited significant accumulation during primary bud necrosis (Figure 4d). These findings suggest an active synthesis and metabolism of auxin during primary bud necrosis. The content of trans-zeatin (TZ), dihydrozeatin (DZ), and GA3, which are plant growth regulators involved in cell division, significantly increased during PBN (Figure 4e,f). Although ethylene levels cannot be directly detected, the precursor, 1-aminocyclopropanecarboxylic acid (ACC), increased five-fold compared to BN, indicating that ethylene may play a significant role in the development of PBN (Figure 4g).

### 2.3. Differentially Expressed Genes during PBN Progression in ‘Summer Black’

Throughout the primary bud necrosis process, A total of 9 RNA-Seq libraries were established at three time points: BN, PN, and AN. High-quality clean data of 64.38 Gb were obtained (Appendix A). Correlation analysis and principal correlation analysis were performed on the sequencing data to check the quality (Figure 5a and Appendix A). The results revealed that PN_1 had higher correlation with three samples in the AN group with R^2^ of 0.975, 0.969 and 0.976, but a low correlation to the other two PN group samples, PN_2 and PN_3, with R^2^ of 0.917 and 0.96, respectively. Additionally, BN_2 deviated from other two samples in the BN group. PN_1 and BN_2 samples were excluded from the further analysis and the remaining repeat samples in each group had high correlations.

**Figure 4 ijms-24-10410-f004:**
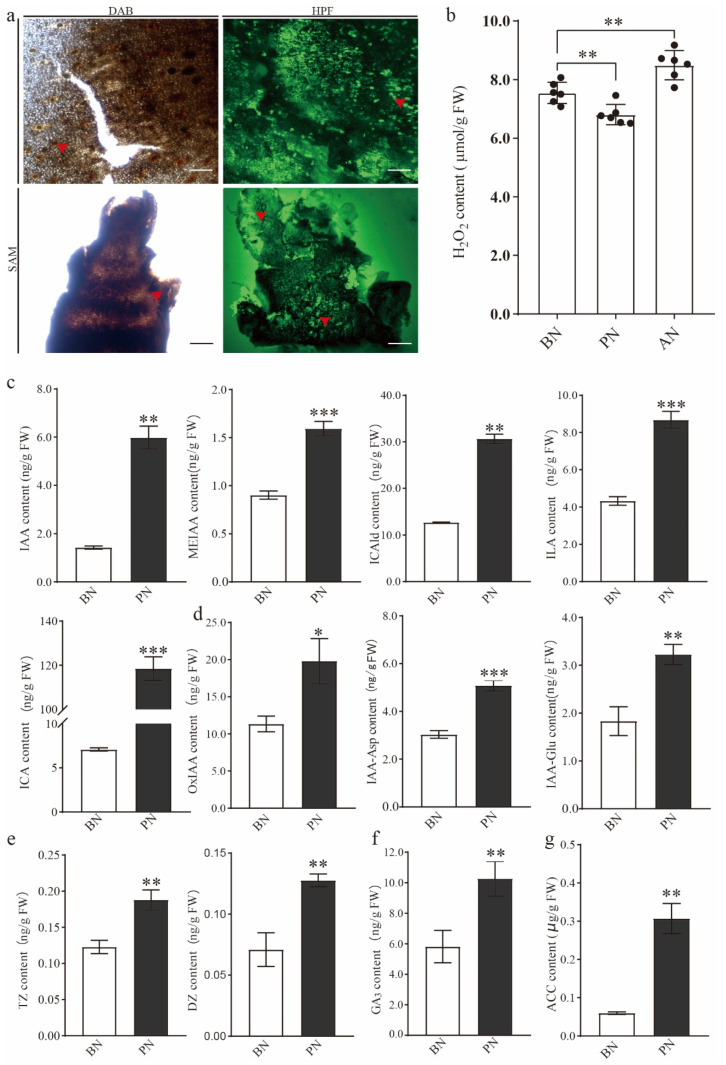
The ROS and phytohormone changes during PBN progression. (**a**) ROS staining. (**b**) H_2_O_2_ content. (**c**) Indolic contents, including IAA, MEIAA, ICA, ICAld, and ILA. (**d**) IAA metabolites contents, including IAA-Asp, IAA-Glu, and oxIAA. (**e**) Cytokinin content. (**f**) GA_3_ content. (**g**) ACC content. ACC, 1-aminocyclopropanecarboxylic acid; DZ, dihydrozeatin; IAA,0 indole-3-acetic acid; IAA-Asp, indole-3-acetyl-L-aspartic acid; IAA-Glu, indole-3-acetyl glutamic acid; ICA, indole-3-carboxylic acid; ICAld, indole-3-carboxaldehyde; ILA, indole-3-lactic acid; MEIAA, methyl indole-3-acetate; oxIAA, 2-oxindole-3-acetic acid; tZ, trans-Zeatin. Data are herein provided as the means ± standard deviation of the mean (n = 3) and were analyzed with Student’s *t*-test. Statistically significant differences are indicated with asterisks (* *p* < 0.05, ** *p* < 0.01, *** *p* < 0.001).

**Figure 5 ijms-24-10410-f005:**
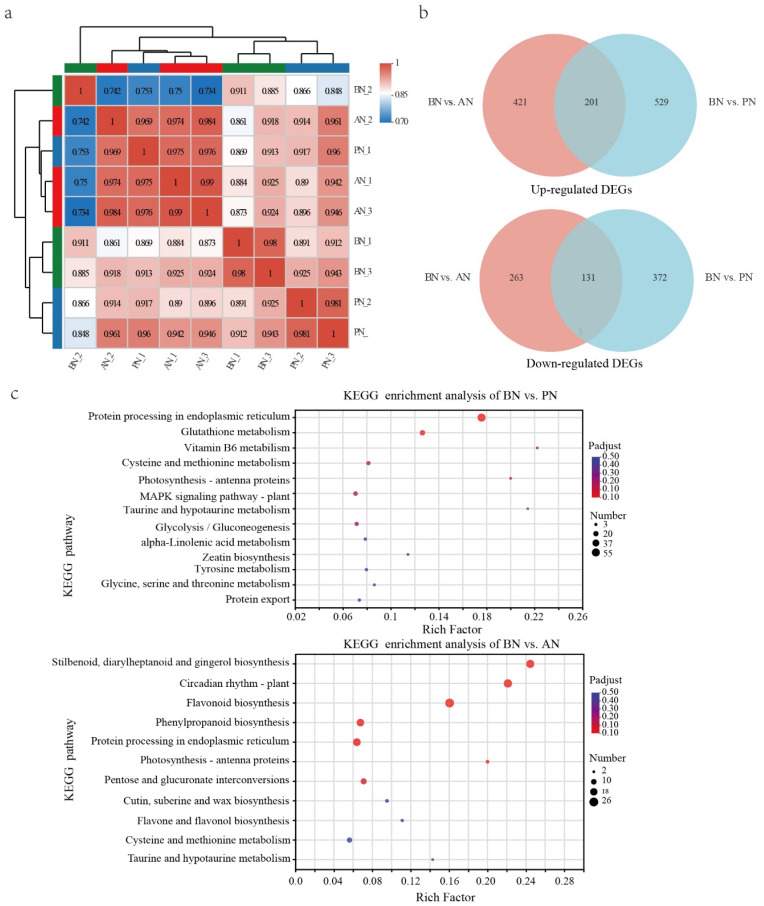
Differentially expressed genes during primary bud necrosis progression. (**a**) Transcriptome correlation evaluation. (**b**) Venn diagram analysis of DEGs from the pairwise comparisons of BN vs. PN and BN vs. AN. (**c**) The KEGG-enriched pathway of DEGs in BN vs. PN and BN vs. AN.

In the comparison of BN vs. PN and BN vs. AN, 1233 and 1016 DEGs were identified, respectively, with 730 upregulated and 503 downregulated in BN vs. PN and 622 upregulated and 394 downregulated in BN vs. AN (Appendix A). The Venn diagram illustrates that 201 genes were upregulated and 131 genes were downregulated in both comparison groups (Figure 5b). Gene ontology (GO) functional classification and enrichment analysis revealed that 1031 and 839 annotated DEGs were enriched to 293 and 461 terms, respectively (Appendix A). In BN vs. PN, the top 20 enriched terms were classified into two categories: stress response (including response to hydrogen peroxide, reactive oxygen species, heat, high light intensity, and glutathione metabolism) and protein processing (including protein self-association, protein complex oligomerization, protein folding, unfolded protein binding, chaperone-mediated protein folding, chaperone cofactor-dependent protein refolding, ‘de novo’ protein folding, de novo posttranslational protein folding, and heat-shock protein binding) (Appendix A). In contrast, the comparison of BN vs. AN shows enrichment of genes that regulate cell growth, reproduction, division, cell cycle, and other cellular processes, apart from stress responses and protein processing (Appendix A).

The Kyoto Encyclopedia of Genes and Genomes (KEGG) enrichment analysis of DEGs showed that protein processing in endoplasmic reticulum, photosynthesis–antenna proteins, MAPK signaling pathway–plant, and glutathione metabolism pathways were significantly different between BN and PN (Figure 5c, Appendix A). Based on the observed cellular stress response (Figure 3), it can be inferred that the pathways related to the stress responses of mitochondria, chloroplasts, and endoplasmic reticulum are involved (Appendix A). Specifically, upregulation of ascorbate peroxidase (*APX*) and glutathione S-transferase (*GST*) genes indicates a stress response in mitochondria, upregulation of calreticulin gene (*CALR*) expression in the endoplasmic reticulum is involved in correctly folding proteins, while ER stress induces upregulation of numerous molecular chaperones, such as heat-shock protein-coding genes (*HSPs*), including *HSP20*, *HSP70*, and *HSP90*, as well as HSP70-binding protein 1 gene (*HSPBP1*), which participates in ER-associated degradation (ERAD). Additionally, ubiquitin-conjugating enzyme gene E2-17 (*UBE2D*) and E3 ubiquitin-protein ligase gene (*CHIP*) may participate in the formation of ubiquitin ligase complexes and ubiquitin-mediated proteolysis, while luminal-binding protein 4 (BIP) is involved in the physiological process of protein recognition by luminal chaperones. In chloroplasts, genes encoding light-harvesting complex chlorophyll a/b binding proteins (LHC), including *LHCA1*, *LHCA4*, *LHCB5*, and *LHCB4,* are upregulated.

Significant differences were also identified in pathways related to secondary metabolism, such as vitamin B6 metabolism, flavonoid biosynthesis, and phenylpropanoid biosynthesis. These findings suggest that the process of PBN involves complex and extensive biological activities.

### 2.4. Differentially Accumulated Metabolites during PBN Progression in ‘Summer Black’

To obtain information on metabolite changes during the PBN progression, targeted metabolomics analysis was performed on primary and secondary metabolites extracted from winter buds of grapevine cv. Summer Black at 60, 67, and 70 DABB. A total of 832 metabolites were detected and grouped into 11 major classes, with flavonoids (228), phenolic acids (142), and lipids (114) being the most abundant classes (Appendix A). The principal component analysis (PCA) revealed that the QC samples exhibited similar metabolic features, indicating high stability and reliability of the overall data analysis. Additionally, the winter bud samples were differentiated by both the PBN degree and temporal changes in the PC2, with a variation of 27.79% (Figure 6a). 

A total of 137 DAMs were detected in BN vs. PN, while 209 DAMs were identified in BN vs. AN. The criteria used for identification were VIP ≥ 1 and fold change ≥ 2 or fold change ≤ 0.5 (Figure 6c and Appendix A). Further comparative analysis revealed that 121 DAMs (31 downregulated and 90 upregulated) were commonly accumulated metabolites in both comparisons (Figure 6b). These DAMs are considered to play a crucial role in understanding the physiological changes associated with PBN. They comprise 35 lipids, 27 others (including sugars and stilbenes), 17 flavonoids, 13 phenolic acids, 7 alkaloids, 7 terpenes, 7 amino acids and derivatives, 4 organic acids, 3 lignans and coumarins, and 1 nucleotide and derivative (Appendix A). There was an overall upregulation trend for lipids, sugars, and terpenes, while flavonoids exhibited an overall downregulation trend (Appendix A). Among the top 20 most differentially accumulated metabolites, 16 DAMs were found to be common to both comparisons, including 12 stilbenes, such as Piceatannol-3′-*O*-glucoside, Tetrahydroxy-stilbene-*O*-glucoside, etc. (Appendix A). It indicated that the onset of necrosis in the primary bud leads to the substantial generation of stilbenes. The most differentially accumulated terpenoids, lipids, lignans, and coumarins were 30-Norhederagenin, LysoPE 17:1 (2n isomer), and ehydrodiisoeugenol. Ascorbic acid is depleted during PN.

The annotated DAMs identified between BN vs. PN and BN vs. AN were mapped to 48 and 55 metabolic pathways, respectively, using KEGG (Appendix A). Among the top 20 enriched pathways, there were some common pathways, including fructose and mannose metabolism, pentose and glucuronate interconversions, arginine and proline metabolism, stilbenoid, diarylheptanoid and gingerol biosynthesis, monobactam biosynthesis, folate biosynthesis, alpha-linolenic acid metabolism, and linoleic acid metabolism (Figure 6d). BN vs. PN comparison had unique pathways, such as plant hormone signal transduction, ABC transporters, arginine biosynthesis, and glutathione metabolism. On the other hand, BN vs. AN comparison had unique pathways, such as biosynthesis of unsaturated fatty acids, pentose phosphate pathway, nicotinate and nicotinamide metabolism, cutin, suberine, and wax biosynthesis.

**Figure 6 ijms-24-10410-f006:**
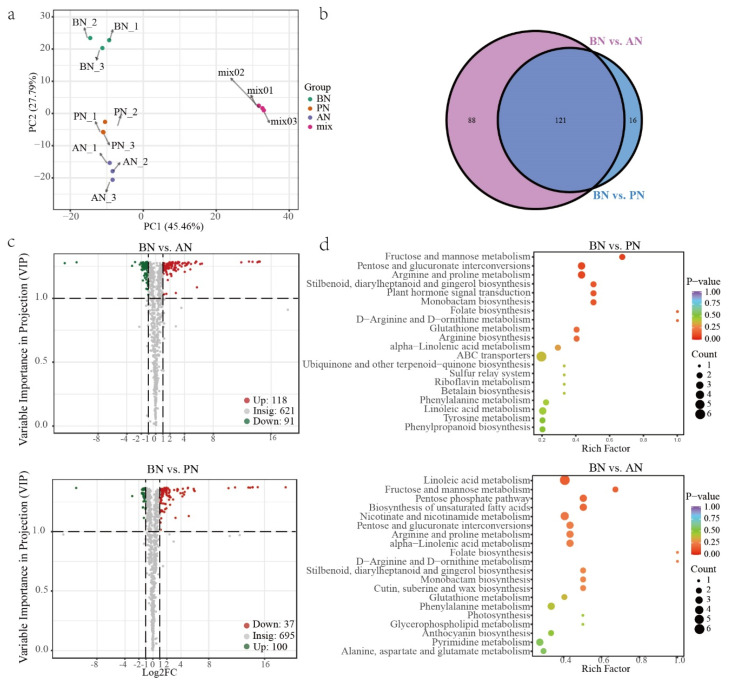
Differentially accumulated metabolites during primary bud necrosis progression. (**a**) Principal component analysis of metabolite profiles. (**b**) Venn diagram analysis and (**c**) volcano plots of differentially accumulated metabolites in comparisons groups of BN vs. PN and BN vs. AN. (**d**) KEGG enrichment analysis of pathway of differentially accumulated metabolites.

### 2.5. The Correlation between DAMs and DEGs during PBN Progression in ‘Summer Black’

Significantly co-enriched pathways were screened based on a Pearson correlation coefficients (PCCs) greater than 0.80 and corresponding *p*-values (PCCP) less than 0.05, and multiple pathways exhibited strong positive correlations between their respective genes and metabolites (Appendix A). The correlation network analysis of 31 DEGs and 20 DAMs on the significantly co-enriched pathways (*p*-value < 0.05) obtained a clue in the secondary metabolism changes during bud necrosis process, involving pathways, such as phenylpropanoid biosynthesis, phenylalanine metabolism, flavonoid biosynthesis, tyrosine metabolism and stilbenoid, diarylheptanoid and gingerol biosynthesis, glutathione metabolism pathway, pentose and glucuronate interconversions pathway, and cysteine and methionine metabolism pathway (Appendix A). The results showed that phenylalanine ammonia-lyase (*PAL*, *Vitvi13g00622*) and shikimate O-hydroxycinnamoyltransferase (*HCT*, *Vitvi11g00720*) are crucial genes involved in the regulation of secondary metabolism. Following primary bud necrosis, there was a decrease in the levels of L-phenylalanine, ferulic acid in the phenylpropanoid biosynthesis pathway, 2-phenylethanol and 2-phenylethylamine in the phenylalanine metabolism pathway, as well as epigallocatechin, gallocatechin, 3,5,7-trihydroxyflavanone, 2′,4,4′,6′-tetrahydroxychalcone, and isosalipurposide in the flavonoid biosynthesis pathway. Conversely, there was an increase in the levels of pterostilbene and piceatannol in the stilbenoid, diarylheptanoid, and gingerol biosynthesis pathway (Figure 7).

When the primary bud underwent necrosis, the gene encoding phenylalanine ammonia-lyase (*VvPAL*, *Vitvi13g00622*) was upregulated and negatively correlated with L-phenylalanine, ferulic acid, 2-phenylethanol, and 2-phenylethylamine. Similarly, the gene encoding shikimate O-hydroxycinnamoyl transferase (*VvHCT*, *Vitvi01g01514*, *Vitvi11g00720*) was downregulated and positively correlated with L-phenylalanine, ferulic acid, epigallocatechin, gallocatechin, 3,5,7-trihydroxyflavanone, naringenin chalcone, 2′,4,4′,6′-tetrahydroxychalcone, and isosalipurposide but negatively correlated with pterostilbene and piceatannol. These results suggest a shift in the secondary metabolic pathway from flavonoid synthesis to stilbenoid biosynthesis. Furthermore, the tyrosine metabolism pathway showed an increase in the content of 3,4-dihydroxybenzeneacetic acid and 3,4-dihydroxy-L-phenylalanine (L-Dopa), and polyphenol oxidase (*VvPPO*, *Vitvi10g00442*, *Vitvi10g02337*, *Vitvi10g02336*) was upregulated, which may be the underlying cause of primary bud browning and necrosis. 

### 2.6. In Situ Expression of VvP23 during PBN Progression in ‘Summer Black’

To investigate key players in the PBN regulatory network, we analyzed the DEGs and compared them with expression characteristics in the floral bud differentiation RNA-seq libraries of grapevine cv. Summer Black (unpublished data). Our results show that *VvHSP90s* were upregulated during bud necrosis (Appendix A). Interestingly, we found that the expression level of *VvHSP90s* was highest in the second nodes, lower in the fourth nodes, and lowest in the sixth nodes at 65 DABB. Meanwhile, the sixth-node buds were undergoing necrosis, whereas the fourth nodes had a relatively lower necrosis rate and severity, and the second nodes had not yet undergone necrosis (more than one week later than the sixth node bud necrosis). The gene encoding the co-chaperone protein VvP23 exhibited constant expression levels before necrosis but exhibited an upregulation in expression during necrosis at 65 DABB, and the expression levels of *VvP23* at the second, fourth, and sixth nodes were similar (Figure 8a). It has been reported that P23, as a conserved and essential co-chaperone of HSP90, participates in the correct folding of client proteins [28,29]. Additionally, P23 interacts with auxin transporters PIN1, PIN3, and PIN7, manipulating the distribution of auxin in the root meristem and controlling root development in *Arabidopsis thaliana* [30,31]. HSP90 and P23 exhibit distinct expression patterns in PBN, with a positive correlation observed between the node position and the severity of necrosis in higher nodes with HSP90 expression, and a negative correlation observed between the expression level of HSP90 and higher expression levels in lower nodes. These findings suggest that VvP23 may have other functions during the process of bud necrosis. Therefore, it is suggested that *VvP23*, the grape homolog of *AtP23*, may participate in regulating the level and gradient of auxin concentration during bud development and play an important role in regulating PBN.

The expression pattern of *VvP23* was investigated using in situ hybridization in the sixth node of grapevine cv. Summer Black during primary bud development and necrosis. The results showed that transcripts of *VvP23* were specifically accumulated in the apical meristematic tissue of the primary bud, with a gradual decrease from the L1 and L2 layers to the L3 layer at 50 DABB (Figure 8b). The expression of *VvP23* was highly accumulated in the branch primordia of the inflorescence primordia at 57 DABB (Figure 8c) but decreased at 63 DABB, in the early stages of primary bud necrosis, and was mainly expressed at the top of the cone-shaped inflorescence primordia (Figure 8d). *VvP23* was significantly upregulated around the inflorescence at 70 DABB, at the time of peak severity of primary bud necrosis (Figure 8e). In summary, in situ hybridization confirmed an increase in the expression of *VvP23* after the occurrence of bud necrosis, with low expression levels along the bud axis and at the junction of the bud base and the bud cushion layer.

**Figure 8 ijms-24-10410-f008:**
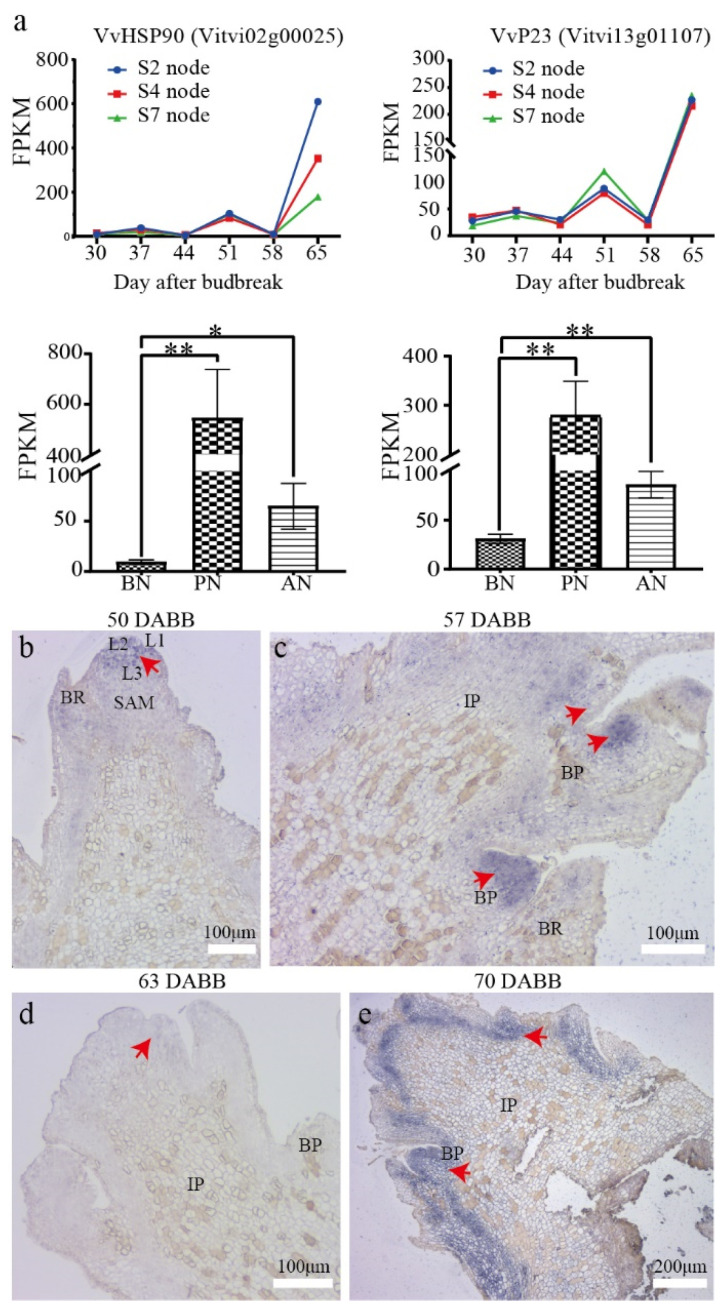
Expression of *VvHSP90* and *VvP23* in ‘Summer Black’. (**a**) Comparative expression analysis of *VvP23*. The FPKM values were analyzed using a two-tailed unpaired *t*-test. Significance levels are denoted as * and **, representing *p* < 0.05 and *p* < 0.01, respectively. (**b**–**e**) In situ hybridization of *VvP23* in 6th nodes of grapevine cv. Summer Black at 50, 57, 63, 70 DABB. The red arrow indicates the probe labels.

## 3. Discussion

### 3.1. The Development Process and Irreversibility of Primary Bud Necrosis

Cell breakdown is a significant feature of primary bud necrosis. The initial cell necrosis occurred in a random, punctate distribution pattern, and little cell breakdown was observed in healthy primary buds (Figure 1), as reported in previous studies [5,8]. Under TEM observation, more severe damage was observed in organelles, such as mitochondria, chloroplasts, the Golgi apparatus, and ER structures. 

The plasmolysis and mitochondria swelling are the most important features of plant cell necrosis that distinguish it from other types of cell death [32]. These features were observed in tissues from various parts of the bud, with the most obvious feature observed in the bracts (Figure 3). The cells in the bracts exhibited significant deformations, indicating a complete disruption of the cell structure. In addition, the cell wall structure was abnormal, with destruction of fiber structures, increased cell wall thickness, and decreased permeability [5,8]. The observation revealed that necrosis may spread through proximity, as cells closely adjacent to the necrotic cells displayed significant membrane separation and expansion of the endoplasmic reticulum into vesicular structures, which may be closely related to the involvement of ROS in cell death.

This study was the first to discover the pattern of the visible symptoms associated with PBN, which involves cell necrosis occurring in the bud axis above the bud cushion and then spreading along the axis towards the top (Figure 2). The apical meristem is the last tissue to undergo cell death. There is a clear physiological boundary that exists between the bud cushion and the bud base. Cell necrosis occurred in the meristems above the bud base, resulting in cell breakdown and the formation of a hollow cavity at the bud cushion. Therefore, the bud base did not undergo necrosis, and bud necrosis ceased at the boundary. This pattern of necrosis suggests that once bud necrosis initiates, it is irreversible. 

### 3.2. The Crucial Role of ROS in Primary Bud Necrosis

ROS are inevitable by-products of the electron transport chains in aerobic metabolism, and they can act as signaling molecules or cause oxidative damage. Numerous reports have identified the involvement of ROS in cell necrosis, and recent research suggests that the necrosis process might be tightly regulated [33]. Plant cells produce ROS in several subcellular compartments, including chloroplasts and mitochondria, particularly under stress conditions [25,34]. An accumulation of ROS was detected in necrotic buds (Figure 4a), inducing the mitochondrial stress response (Figure 3). This accumulation of ROS resulted in the depletion of non-enzymatic antioxidant systems and the upregulation of the enzymatic antioxidant system, particularly the glutathione metabolism pathway, such as *VvGST* and *VvAPX* (Figure 5c, Appendix A). 

ROS signaling triggers a cascade reaction involved in primary bud necrosis. Mitochondria are extremely sensitive organelles that can be easily damaged. ROS, hypoxia, and changes in osmotic pressure can cause mitochondrial swelling, which can sometimes be a manifestation of increased function, while more noticeable swelling is a sign of cellular damage [26]. Long-term stress or severe damage can disrupt intracellular ROS homeostasis, induce activation of the MAPK signaling pathway, mitochondrial swelling, or dysfunction, preventing cells from activating apoptotic pathways, and cause mitochondrial swelling rather than shrinkage [35,36]. This results in an inability to eliminate or replenish damaged mitochondria, leading to irreversible cell injury, depletion of ATP, and, ultimately, cell death [37,38]. The mechanism of necrotic cell death is conserved across eukaryotes [27]. 

ROS homeostasis was also disrupted in chloroplasts, leading to oxidative damage. LHC proteins function in light harvesting and photoprotection in chloroplasts, and under stress conditions, the expression of LHC is typically repressed. However, a study conducted on spinach found that ROS induces oxidative modifications in amino acid residues in a subset of LHC proteins [39]. It is, thus, inferred that the upregulation of *VvLhc* expression could be a result of VvLHC protein damage. In addition, ROS is closely associated with lipid peroxidation reactions [40]. ROS can react with phospholipids and enzymes in the plasma membrane, leading to an increase in lipid peroxidation metabolic products and triterpenoids. The abundant accumulation of products of polyunsaturated fatty acids (PUFAs) in the metabolome reflects the extent of lipid damage caused by lipid peroxidation. Lipid peroxidation can cause increased membrane permeability, disruption of lipid metabolism in the membrane, modification of protein–DNA covalent bonds, and cell death. This process is similar to ferroptosis, where PUFAs drive ferroptosis [40,41].

### 3.3. Protein Misfolding and Aggregation Are Closely Related to Primary Bud Necrosis

Protein misfolding and aggregation have been associated with numerous protein conformational diseases, such as neuromuscular degeneration and Parkinson’s disease [42,43]. The ER is the main site for protein synthesis and transport, and stresses induce a certain amount of unfolded protein that is toxic to cells. The accumulation of misfolded proteins disrupts ER homeostasis, leading to ER stress [44]. 

During the process of primary bud necrosis, chromatin condensation and protein aggregation were observed using TEM (Figure 3c). Additionally, a large number of DEGs were found to be associated with protein folding and enriched in protein processing pathways in the ER (Figure 5c, Appendix A), including ERAD, protein recognition by luminal chaperones, and the ubiquitin ligase complex. These pathways function as protein quality control systems to remove non-folded proteins and maintain ER homeostasis [44]. Necrotic signaling leads to mitochondrial dysfunction and protein homeostasis imbalance, resulting in an enhancement of the molecular chaperone protein system and two protein hydrolysis systems to maintain protein homeostasis [45,46]. Mitochondrial dysfunction affects the degradation of aggregated proteins mediated by mitochondria [47]. Therefore, the mechanism of bud necrosis may be similar to that of related diseases in medical research. In the process of PBN, molecular chaperones, such as HSPs, may play an important role in maintaining protein stability during stress conditions and suppressing cell apoptosis but also function as signal protein molecular partners and directly promote cell apoptosis [48].

### 3.4. Tissue Browning Is a Significant Characteristic of the Primary Bud Necrosis

Tissue browning is a significant characteristic of plant damage and cell death, closely associated with ROS [49]. It is a noticeable effect observed during the process of PBN (Figure 1). Browning includes enzymatic browning and non-enzymatic browning. Non-enzymatic browning is mainly mediated by the Maillard reaction, such as oxidative discoloration of phenols and ascorbic acid browning [50,51]. Enzymatic browning, on the other hand, is mainly related to PAL, PPO, and peroxidase (POD). These enzymes catalyze the oxidation of phenolic compounds, leading to the formation of quinones, which further react to produce brown pigments [52].

The transcriptome analysis revealed that *VvPAL*, *VvPPO*, *VvTAT*, and *VvPOD* were significantly upregulated (Figure 7, Appendix A). VvPPO is the main enzyme involved in enzymatic browning, which catalyzes the substrates derived from phenylalanine (such as anthocyanins, flavonoids, chalcones, and xanthones) to undergo enzymatic browning in the cytoplasm following disruption of the membrane structure [51,53]. 

Metabolomic analysis showed a decrease in flavonoids, while lipids and stilbene showed an increasing trend (Figure 7). Stilbene and flavonoids belong to polyketides that are produced through the acetylation of p-coumaric acid in the phenylalanine metabolic pathway. Stilbene and flavonoids are polyketides produced by acetylation of p-coumaric acid in the phenylalanine metabolism pathway. Chalcone synthase (VvCHS) and stilbene synthase (VvSTS) have highly similar sequence structures and share the same substrates. The transcriptional regulation controls the carbon flux between these two opposing metabolic pathways [54]. The majority of *VvSTS* is located on chromosome 16, upregulated during primary bud necrosis (Appendix A) [54]. The flavonoid hydroxylase VvCYP75A, along with CYP73A of the cytochrome P450 family, may participate in the hydroxylation of flavonoids, catalyzing the hydroxylation of flavonoid B ring 5′ to enhance tissue antioxidant defense and resulting in a significant decrease in flavonoid content [55,56].

### 3.5. The Role of Endogenous Hormones in Primary Bud Necrosis

Phytohormones play a crucial role in regulating plant life activities. Through hormonal signaling, plants can rapidly adjust their growth and structure to adapt to changing environmental conditions [57]. In this study, we observed the involvement of hormones in bud necrosis. Programmed necrosis and programmed cell death (PCD) are strategies adopted by plants during growth, development, and stress responses, with necrosis being a substitute for PCD [58].

Interestingly, the detection of hormones revealed a significant increase in all tested hormone types (Figure 4c–g). Among them, ethylene has been demonstrated to play a role in PCD in various species, which includes mediating chlorophyll degradation and leaf senescence [59,60]. The experimental results showed that the ethylene precursor ACC content in PN was five-times higher than in BN, indicating a direct association between ethylene and the occurrence of bud necrosis. Auxin plays a role in cell growth and maintains the activity of meristematic cells while antagonizing cell death. The content of IAA and other indolic derivatives significantly increased in PN, particularly the content of indole-3-carboxylic acid (ICA), which increased by 20-times. The increased levels of indolic compounds indicate enhanced growth activity in the necrotic tissues. It has been suggested that indole derivatives are prerequisite substances for plants to resist necrotic pathogens, achieving stress resistance through the plant hormone signaling pathway [61]. An enhanced activity of IAA oxidation through the GH3-ILR1-DAO pathway was also observed, as evidenced by the significant increase in oxIAA, IAA-Asp, and IAA-Glu along this pathway. IAA-Asp and IAA-Glu are considered precursors for auxin degradation, and oxIAA is the product of irreversible oxidation of IAA-Asp and IAA-Glu [62,63]. The co-chaperone P23 has been identified as participating in auxin redistribution and root growth in plants [30]. Here, the grapevine homolog gene *VvP23* was observed to be distributed in the apical meristem and branch primordia, indicating its potential role in the redistribution of auxin and its ability to counteract tissue necrosis. When necrosis occurs, the auxin content increases, and VvP23 participates in auxin transport and redistribution, leading to the formation of structural growth hormone boundaries. The increase in cytokinin and gibberellin content provides evidence that cells resist necrosis through growth antagonism. However, some studies have shown that auxin participates in inducing PCD by stimulating ethylene synthesis [61].

## 4. Materials and Methods

### 4.1. Plant Material

The experiments were conducted during four successive seasons from 2018 to 2021 at ‘Ganshan’ vineyard of Hunan Agricultural University (Hunan, China) (28°08′ N, 113°10′ E). Nine-year-old table grapes ‘Summer Black’ (hybrid of *V. vinifera* and *V. labrusca*) under rain shelter cultivation were used as material in the study, which were under V-shaped horizontal shoot positioning training system and subjected to the standard cultural practices commonly used in commercial vineyards. The next experiments included replicates including means shoot counts but not the vine counts. The distance between the shoots was 20 cm, controlled by a rope. The shoots N were tipped from seventh nodes, and the shoots (N + 1) were trimmed above base leaf except the top lateral shoots at 30 DABB. The shoots of all samples in this experiment were selected using stratified random sampling.

For staining observations, shoots were randomly selected and pruned at third-node position. The detached prunings were then placed in a cooler with ice packs and transferred to the lab immediately. The sixth nodes were cut off and dissected, followed by other tests. The samples for other experimental purposes were cut off from the sixth node and plunged into liquid nitrogen in the field and then mixed and divided into small portions, stored at −80 °C for later use.

### 4.2. Determining Bud Necrosis in the Field

To identify whether the primary bud encounters necrosis, a rapid identification method is applied in the field, using a blade to make a transverse cut at the 1/3 position of the bud (above the bud base) and observing it. The buds with browning features indicate undergoing necrosis.

### 4.3. Trypan Blue Staining

Primary buds were dissected into apical primordia and bracts under a dissecting microscope (Olympus SZX16, Tokyo, Japan) [64]. The apical meristem, along with the bud cushion and bud base, was embedded in 6% agarose gel (Transgen, Beijing, China) and then hand-sectioned to a thickness of approximately 100 µm. Bracts and apical meristem sections were stained as previously described [65,66]. Subsequently, the samples were examined and captured using a Zeiss Axio Imager 2 microscope (Carl Zeiss, Jena, Germany).

### 4.4. ROS Staining

Bracts and apical meristem sections were prepared as described above. ROS detection was performed by staining with DAB [67,68] (Sigma-Aldrich, Shanghai, China) and HPF [69,70] (Maokang Bio-technology, Shanghai, China). The samples were examined and captured immediately using a ZEISS Axio Imager 2 microscope (Carl Zeiss, Germany).

### 4.5. TUNEL Assay

The primary buds sampled from 50, 60, 68, to 70 DABB were embedded in paraffin and cut into 8 μm thicknesses (Leica RM2235, Leica, Wetzlar, Germany). TUNEL assays were performed following the instructions provided in the TMR TUNEL Detection Kit (Servicebio, Wuhan, China). A positive control was prepared by section genomic DNA that was digested by the DNase I (20 U/mL) (Servicebio, China), while a negative control was prepared by replacing TdT enzyme with double-distilled water. The sections were examined and captured with fluorescence microscopy (ECLIPSE C1. Nikon, Japan), and fragmented DNA of necrotic cells was fluorescently stained with TMR and defined as TUNEL-positive.

### 4.6. TEM Analysis

The primary buds sampled at 60 DABB and 67 DABB were dissected into apical primordia, bracts, and bud base, and ultrathin sections were counterstained with 3% uranyl acetate and 2.7% lead citrate and observed using an HT7800 TEM (H-7800, Hitachi, Tokyo, Japan).

### 4.7. Detection of Phytohormones

Winter buds collected from 60 DABB and 67 DABB were sent to Metware Biotechnology (Wuhan, China) to detect endogenous hormone concentrations. The content of auxin, cytokinin, and gibberellin was measured using the AB Sciex QTRAP 6500 LC-MS/MS platform with internal standard methods, while the ACC content was determined using the Agilent 7890B-7000D GC-MS/MS platform. Three biological replications were performed.

### 4.8. Determination of H_2_O_2_

The H_2_O_2_ content was performed using H_2_O_2_ assay kits (Grace-bio, Suzhou, China), and statistical analysis was performed using student’s *t*-test (GraphPad, San Diego, CA, USA). There were 6 biological replicates for each of the sampling times at 60, 67, and 70 DABB.

### 4.9. RNA Isolation and Transcriptome Analysis

Winter buds of grapevine cv. Summer Black were sampled at 60, 67, and 70 DABB for RNA sequencing analyses. A total of nine sequencing libraries were constructed by Majorbio Company (Majorbio, Shanghai, China). The obtained clean reads were annotated to *Vitis vinifera* genome (12X.v2, VCost.v3) [71]. DEGs were defined as having an adjusted *p* < 0.05 and |log2FC| ≥ 1 in the comparisons of BN vs. PN and BN vs. AN.

### 4.10. Widely Targeted Metabolomics Assay

Samples for targeted metabolomics analysis were prepared following the same procedure as described for transcriptome analysis. The UPLC-MS/MS system (UPLC, SHIMADZU Nexera X2; MS, Applied Biosystems 4500 Q TRAP) was used for metabolite data acquisition, following standard procedures [72,73]. The identification of metabolites was carried out using the MWDB database (Metware, Wuhan, China), while metabolite quantification was performed using multiple reaction monitoring (MRM) in triple-quadrupole mass spectrometry [74]. DAMs were screened by combining multivariate and univariate statistical analyses with VIP ≥ 1 and (|Log_2_(fold change)| ≥ 1.0, using the R package [75]. The identified DAMs were then annotated with KEGG database [76].

### 4.11. Integrative Analysis of Transcriptome and Metabolome

The transcriptome and metabolome data were log2-transformed, and significant metabolites and related DEGs were screened out within the joint correlation analysis, with a threshold of PCC > 0.80 and PCCP < 0.05 and mapping to KEGG pathways [77,78].

### 4.12. In Situ Hybridization

Samples of primary buds collected at 50, 57, 63, and 70 DABB were processed into 8 μm thick sections, and hybridization was detected using NBT-BCIP following the protocol described in a previous study [79]. The antisense RNA probe was synthesized by the solid-phase phosphoramidite triester method (Servicebio, Wuhan) and labeled with digoxigenin. The probe sequence was 5′-DIG-CATAGACGATGCTTCTGACTCCAACATTACACT-3′.

## Figures and Tables

**Figure 2 ijms-24-10410-f002:**
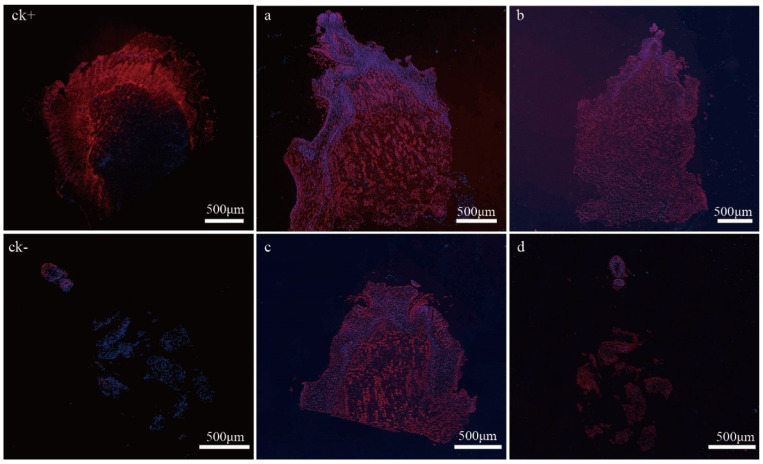
Detection of genomic DNA degradation and primary bud necrosis development via TUNEL assay. (**a**–**d**) TUNEL detection of primary bud necrosis at 50, 60, 67, 70 DABB; ck+ is a positive control with genomic DNA digested by the DNase Ⅰ, ck– is a negative control without TdT.

**Figure 7 ijms-24-10410-f007:**
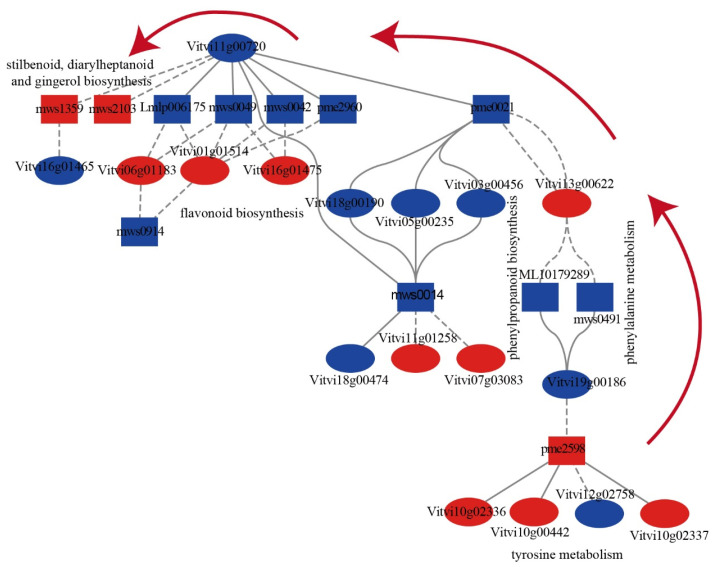
The relationship between the differentially abundant metabolites and differentially expressed genes during primary bud necrosis progression. Correlation network diagrams of DEGs and DAMs enriched in phenylpropanoid biosynthesis, phenylalanine metabolism, flavonoid biosynthesis, tyrosine metabolism, and stilbenoid, diarylheptanoid, and gingerol biosynthesis pathways; the ellipse represents the transcript node and rectangle represents the metabolite node. Red and blue correspond to increases and decreases in genes expression and metabolite abundance, respectively. Positive and negative correlations are presented as solid line and dash line, respectively. ML10179289, 2-phenylethanol; mws0491, 2-phenylethylamine; pme2598, 3,4-dihydroxybenzeneacetic acid; mws0914, 3,5,7-trihydroxyflavanone; mws0042, epigallocatechin; mws0014, ferulic acid; mws0049, gallocatechin; Lmlp006175, isosalipurposide; pme0021, L-phenylalanine; pme2960, naringenin chalcone; mws2103, piceatannol; mws1359, pterostilbene.

## Data Availability

All relevant data and figures in this study can be found within the article and its Appendix A.

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
