# Peer review of "Integrated Transcriptome and Metabolome Analysis Revealed the Causal Agent of Primary Bud Necrosis in ‘Summer Black’ Grape"

_ijms, 2023, doi:10.3390/ijms241210410_

Round 1

Reviewer 1 Report

The present manuscript entitled ‘Overexpression of the poplar WRKY51 transcription factor enhances salt tolerance in Arabidopsis thaliana’ written well. However, authors are advised to read the manuscript carefully and correct the language accordingly. In some of the sentences capital letters are used unnecessarily. Apart from this, gene names should be in italic in whole manuscript. Under Conclusion gene names are written in normal.

See above comments

Author Response

I'm sorry, reviewer, you might made an easy mistake about the review comments. The title of the submission you mentioned is "Integrated transcriptome and metabolome analysis revealed the causal agent of primary bud necrosis in 'Summer Black' grape". We appreciate your time and are eagerly awaiting your comments. Thank you.

Reviewer 2 Report

The article „Integrated transcriptome and metabolome analysis revealed causal agent of primary bud necrosis in ‘Summer Black’ grape” is a comprehensive, detailed and interesting topic.

The structure of article is: Introduction, Results, Discussion, Materials and Methods, References and Supplementary materials.

The aim of paper was to investigate the development and severity of PBN in grapevine cv. Summer Black, and revealing the underlying mechanism through employing transcriptomics and metabolomics technologies.

Results consist of: Primary bud necrosis progression in ‘Summer Black’, The changes in hormone and H2O2 levels during PBN progression in ‘Summer Black, differentially expressed genes during PBN progression in ‘Summer Black’, Differentially accumulated metabolites during PBN progression in ‘Summer Black, The correlation between DAMs and DEGs during PBN progression in ‘Summer Black, In situ expression of VvP23 during PBN progression in ‘Summer Black.

 Discussion consists of following parts: The development process and irreversibility of primary bud necrosis, The crucial role of ROS in primary bud necrosis, Protein misfolding and aggregation are closely related to primary bud necrosis, Tissue browning is a significant characteristic of the primary bud necrosis. The role of endogenous hormones in primary bud necrosis. In the discussion mechanisms of action are well explained and supported by adequate literature.

Materials and Methods consists of: Plant Material, Determining bud necrosis in the field, Trypan blue straining, ROS staining, TUNEL assay, TEM analysis, Detection of phytohormones, Determination of H2O2, RNA isolation and transcriptome analysis, Widely targeted metabolomics assay, Integrative analysis of transcriptome and metabolome, In situ hybridization.

In this article Pearson correlation between respective genes and metabolites was used to support presented data.

References consist of 78 literature which correspond the topic of the paper.

This article may be accepted for publication in this journal after minor revision.

Suggested corrections:

Line 62 - 68 which is correlation pear and Summer Black’ grape?  better explain

Check English.

Minor corrections are needed

Author Response

Dear reviewer, We have revised the manuscript by addressing the suggested corrections you provided. Thank you for your valuable evaluation. Please refer to the attached document for our detailed response. Best regards, Shaogang Fan

Reviewer 3 Report

Review Report

General comments: -

The topic is very interesting and has a great impact on the field. The manuscript is fairly written and suitable to be published in the IJMS after taking care of some MINOR comments.

Detailed comments: -

In general, please avoid using the personal pronouns (I, We, our) as it was found line 11 (We Discovered), Line 383: We also found and more. Please apply this rule throughout the manuscript.

Keywords:

Please add the word transcriptome to the keywords list.

Abstract:

The aim is not clearly stated in this section. Please state the aim or the objectives clear and specific.

It is better to provide some values in this section.

Introduction:

This section needs to be enriched and expand by adding more background about the topic.

Materials and Methods:

The experimental design is adequate and suitable to the current study.

Results

This section is ok BUT the data presentation needs some modifications for better understanding.

For Example:

Figure 2: Tunnel Assay needs to be thoroughly discussed.

-Figure 4c Auxin and Indole derivatives contents needs to be divided into two panels as C for Auxin derivatives contents and D for Indole derivatives therefore Figure 4 will include 7 panels(A-G).

-Also Figure 5 Differentially Expressed genes during primary bud necrosis progression

For panel b, I strongly suggest the author to put the down regulated and up-regulated DEGs in one diagram with different color code for better illustration and presentation.

Discussion:

This section is poorly written.

THE AUTHOR IS ADVISED TO REWROTE THIS SECTION WITH MORE ATTENTION. DISCUSSION OF THE DATA IN FIGURES IS TOTALLY missing.

PLEASE WHEN YOU WRITE THE DISCUSSION SECTION RELATE AND RELY ON RESULTS AND DATA PROVIDED IN EACH CORRESPONDING FIGURE WITH A COMPLETE INTEPRETATION.

Conclusion:

This section is not required but it will be better to add this section to include the study and provides some significant data with some recommendations for future work.

References:

The authors provided enough citations but missing some for a stronger background.

It is UPTODATE.

English Language and style is ok. Some minor editing is required.

Author Response

Dear reviewer, We would like to express our gratitude for your valuable feedback and suggestions. We have carefully reviewed the manuscript and have made revisions to the Results section, and modified the Discussion section to improve its clarity and coherence. During the revision process, we have given careful consideration to the wording, sentence structure, and overall flow of the manuscript, aiming to enhance its readability and scientific quality. We sincerely appreciate your time and effort in reviewing our manuscript and providing us with constructive feedback. Your input has been instrumental in improving the quality of our work. Thank you once again for your valuable contribution. Thank you for your valuable evaluation. Best regards, Shaogang Fan

Reviewer 4 Report

The authors investigated the development and severity of Primary bud necrosis (PBN) physiological disorder, in grapevine, that leads to decreased berry yield. To understand the regulatory process of PBN development and reveal the underlying mechanism, different parameters were evaluated, using transcriptomics and metabolomics technologies.

The article is clearly written, well-structured and the references are recent and adequate. This study provides important clues for further study on PNB. The results found are very interesting and the article can be accepted in the present form.

Minor changes:

Line 165 – delete the comma and insert a period.

Line 230 - insert a space before the parentheses “genes (CALR)”.

Line 292 – delete one of the parentheses.

Line 596 - delete a dash.

Line 602 – “L.” no italics.

Line 618 - abbreviate journal name.

Line 624 - complete the page number.

Line 708 - delete the comma and insert a period “Wood R M.   ”

Line 725 - “L.” no italics.

Line 726 - delete the comma and insert a period “Tang G.   ”

Author Response

Dear reviewer, We have revised the manuscript by addressing the suggested corrections you provided. Thank you for your careful corrections and valuable evaluation. Best regards, Shaogang Fan

Reviewer 5 Report

General Comments:

In this manuscript entitled “Integrated transcriptome and metabolome analysis revealed causal agent of primary bud necrosis in ‘Summer Black’ grape” authors have explored the physiological changes that occur in primary bud necrosis of grape. Also, authors have correlated those physiological changes by transcriptomics and metabolomics. Experimental approaches are good.  Discussion is poor.

Specific comments:

Line 4: Straining or staining?

Moderate editing of English language required

Author Response

(The authors gave the same response as above.)
